# Reasoning-VLA: An Efficient and Spatial-Guided General Vision-Language-Action Reasoning Model for Autonomous Driving

Dapeng Zhang [1][2][*]   Zhenlong Yuan [3][*]   Zhangquan Chen [4]   Chih-Ting Liao [5]   Yinda Chen [6]   Fei Shen [1]
Qingguo Zhou [2]   Tat-Seng Chua [1]

## Abstract

Vision-Language-Action (VLA) models have recently shown strong decision-making capabilities in autonomous driving. However, existing VLAs often struggle with achieving efficient inference and generalizing to novel autonomous vehicle configurations and driving scenarios. In this paper, we propose Reasoning-VLA, a general and efficient action-generation VLA framework. The proposed model employs a set of learnable action queries, implicitly guided by predefined spatial representations to enhance spatial awareness. These learnable queries interact with reasoning-enhanced vision–language features to generate continuous action trajectories in parallel. To promote robust generalization, we consolidate eight publicly available autonomous driving datasets into a standardized, Chain-of-Thought reasoning–based, and easy-to-use data format for model training. Leveraging both supervised learning and reinforcement learning fine-tuning, extensive empirical evaluations across multiple benchmarks demonstrate that Reasoning-VLA achieves state-of-the-art performance, strong generalization capability, and the excellent inference speed with parallel decode.

## 1. Introduction

Autonomous driving (AD) is a highly complex system that requires precise environmental perception and reliable driving behavior generation. Traditional end-to-end AD methods initially advanced the field but face issues such as poor scalability, cumulative errors, and limited generalization across hardware and datasets. These limitations hinder their generalization ability to new driving scenarios. Recently, foundation models—especially large language and vision–language models like CLIP, Qwen2.5-VL, and DeepSeek-V3 (Radford et al., 2021; Bai et al., 2025; Liu et al., 2025)—have shown remarkable generalization through large-scale pretraining. Their capabilities offer a promising direction for building more flexible and robust AD systems.

Building on these advancements, contemporary frameworks in robotic manipulation and autonomous driving increasingly adopt vision–language generative paradigms (e.g., autoregressive or diffusion-based models (Black et al., 2024; Kim et al., 2024; Yuan et al., 2025; Yang et al., 2025)), collectively referred to as Vision–Language–Action (VLA) models. These systems generate fine-grained action trajectories from high-level visual–linguistic reasoning, thereby enhancing flexibility and practicality in motion planning and control. Leveraging large-scale pretrained foundation models, recent approaches such as DriveMOE (Yang et al., 2025) have achieved strong benchmark performance while simultaneously improving interpretability and robustness capabilities in autonomous driving tasks.

Despite these promising results, **several challenges** hinder the widespread deployment of VLAs in autonomous driving: 1) Most existing VLA architectures are based on autoregressive or diffusion models that require multiple inference steps to generate actions, limiting their suitability for real-time, high-frequency control. 2) Current VLA methods lack robust generalization to new vehicle platforms or unseen driving scenarios. We argue that developing a general-purpose foundation VLA requires diverse, large-scale datasets that encompass various environments and vehicle configurations. 3) Existing fine-tuning strategies are often inefficient in exploring the full potential of VLAs, constraining their generalization capability.

To address these challenges, we propose Reasoning-VLA, an efficient and generalist VLA framework that establishes a new state-of-the-art for autonomous driving. First, we design a novel interaction mechanism between action and

---

[*]Equal contribution  [1]National University of Singapore, Singapore [2]Lanzhou University, China [3]Meituan LongCat Team [4]Tsinghua University, China [5]University of New South Wales, Australia [6]University of Science and Technology of China, China. Correspondence to: Fei Shen <shenfei29@nus.edu.sg>, Qingguo Zhou <zhouqg@lzu.edu.cn>.

*Proceedings of the 43rd International Conference on Machine Learning*, Seoul, South Korea. PMLR 306, 2026. Copyright 2026 by the author(s).

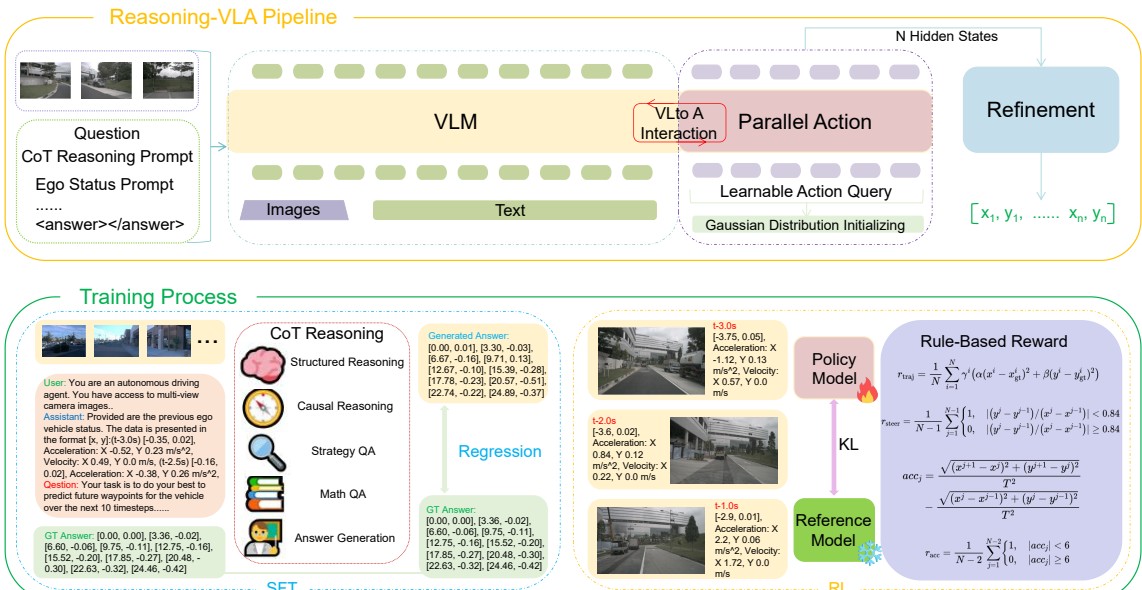

*Figure 1.* Reasoning-VLA is an efficient Vision–Language–Action (VLA) framework for autonomous driving that employs parallel actions to interact with reasoning-enhanced vision–language models (VLMs), enabling one-step prediction of future trajectories. The model is trained on our unified and generalized autonomous driving dataset using a combination of supervised fine-tuning (SFT) and reinforcement learning (RL), guided by specifically designed rule-based reward functions.

vision–language modalities by introducing a set of learnable action queries, implicitly guided by predefined spatial representations to enhance spatial awareness. These learnable queries interact with reasoning-enhanced vision–language representations through cross-attention to extract action-related information efficiently and generate continuous trajectories in parallel. Second, to enable generalization, we construct a unified, Chain-of-Thought reasoning-based dataset that merges eight publicly available autonomous driving datasets into a coherent and easy-to-use format. This dataset covers diverse vehicle platforms and driving scenarios, enhancing the generalization ability of Reasoning-VLA. Finally, we adopt a two-stage training strategy that combines supervised fine-tuning (SFT) and reinforcement learning (RL) to fully exploit the model's reasoning and planning potential. Extensive experiments demonstrate that Reasoning-VLA significantly improves generalization ability, planning performance, and inference speed compared with existing VLA approaches. To summarize, the main contributions are as follows:

- We propose Reasoning-VLA, an efficient and general VLA framework that employs learnable action queries to interact with reasoning-enhanced vision–language representations, enabling one-step parallel action generation.

- We introduce an Implicit Spatial Guidance strategy, which enhances spatial awareness during the initialization of learnable action queries.

- We construct a unified, Chain-of-Thought reasoning-

based autonomous driving dataset that merges eight existing datasets, facilitating generalization across vehicle types and driving environments.

- We employ a combined SFT and RL fine-tuning strategy augmented with physical and dynamic rewards to enhance the general reasoning ability of Reasoning-VLA, achieving substantial improvements over prior methods.

## 2. Related Work

### 2.1. Vision-Language-Action Models

With the rapid advancement of VLMs in recent years (Radford et al., 2021; Zhu et al., 2023; Li et al., 2022a; 2023; Bai et al., 2025; Liu et al., 2025; Zhu et al., 2025), researchers have increasingly integrated VLMs into autonomous driving and robotic systems to enhance overall performance. For instance, DriveVLM and DriveMM (Tian et al., 2024; Huang et al., 2024) incorporate VLM modules to improve situational understanding and enhance generalization in vehicle control. DriveMLM (Wang et al., 2023) introduces a behavior planning module that produces optimal driving decisions along with rationales. Although these methods effectively model vision-language representations, they often neglect the role of action generation, limiting their practical applicability. To address this, recent works have explored integrating vision-language understanding with action prediction, directly fine-tuning large pre-trained VLMs to estimate robot actions (Brohan et al., 2023; Black et al., 2024; Zhang et al., 2025b). These approaches, commonly

referred to as Vision–Language–Action (VLA) models. Recent representative VLA methods demonstrate significant performance improvements. OpenVLA (Kim et al., 2024) employs a pre-trained VLM combined with a discretization bin tokenizer to predict actions. Similarly, $\pi_{0.5}$ (Intelligence et al., 2025) leverages co-training and hybrid multimodal examples—incorporating robot observations, language instructions, and low-level actions—within a single unified model, achieving SOTA performance. The success of VLAs in robotics provides a promising direction for autonomous driving. Some approaches extend novel VLM architectures to train billion-parameter policies with task-specific modifications, offering a direct pathway for AD systems to benefit from rapid VLM advancements (Yuan et al., 2025). However, most existing VLA methods rely on autoregressive or diffusion-based training and inference, which inherently limits their speed and efficiency (Yang et al., 2025).

## 2.2. Fine-tuning with Reinforcement Learning

With the development of extensive pre-training techniques and high-level general capabilities, Reinforcement Learning (RL) has achieved remarkable success in advancing the reasoning and decision-making abilities of LLMs. Reinforcement Learning from Human Feedback (RLHF) approaches, such as PPO (Schulman et al., 2017), typically require training a reward model to optimize the policy network. However, this process can be complex and often unstable. Notably, models such as GPT-4 (OpenAI et al., 2024) follow this RL-based fine-tuning paradigm. Building upon PPO, DPO (Rafailov et al., 2023) fine-tunes pre-trained models to follow instructions and align with human preferences, while eliminating the need for sampling during fine-tuning. Similarly, Qwen3 (Wu et al., 2025) employs DPO to improve performance in applications. Another variant, GRPO (Shao et al., 2024), uses sampling to estimate advantages, thereby effectively enhancing the reasoning capabilities of actors. For example, DeepSeek-R1 (Guo et al., 2025) applies GRPO to advance LLM reasoning, emphasizing self-evolution rather than fine-tuning data. Inspired by these methods, recent works have adopted analogous RL-based fine-tuning strategies to improve the reasoning and decision-making capabilities of autonomous driving models (Yuan et al., 2025).

## 3. Method

As illustrated in Fig. 1, the Reasoning-VLA framework comprises three main components: (1) a reasoning-enhanced vision–language model (VLM) backbone, (2) an action module that interacts with the VLM and enables parallel decoding of action trajectories, and (3) a multi-stage intermediate refinement module. In the following sections, we present a detailed description of our approach to developing a VLA framework for autonomous driving and highlight key insights.

## 3.1. Preliminaries: Vision-Language Models

In this work, we adopt Qwen2.5-VL (Bai et al., 2025) as our foundational model. Qwen2.5-VL effectively simulates human-like analytical thinking, supporting multi-step reasoning, deliberate planning, and problem-solving. Qwen2.5-VL incorporates several architectural innovations: a re-designed Vision Transformer (ViT) with 2D-RoPE and windowed attention for computational efficiency; an MLP-based vision–language merger that compresses visual features into tokens suitable for the LLM; and a large language model initialized with pre-trained Qwen2.5 weights. The model not only exhibits strong vision–language understanding but also maintains robust LLM reasoning capabilities. Furthermore, it generalizes effectively across domains without requiring task-specific fine-tuning, making it a suitable base model for applications such as autonomous driving and action execution in real-world scenarios.

## 3.2. The Structure of Reasoning-VLA

Most existing Vision-Language-Action (VLA) methods either rely on a specialized action tokenizer to convert actions into a format compatible with LLMs, followed by autoregressive generation or employ diffusion/flow matching modules to refine VLM hidden states or noise in order to produce continuous action values. In contrast, our Reasoning-VLA, built on the Qwen2.5-VL symbolic reasoning framework, fundamentally differs from these autoregression-based and diffusion-based approaches (Black et al., 2024; Kim et al., 2024; Yuan et al., 2025). To bridge vision-language representations and action prediction, Reasoning-VLA comprises three primary components: A pre-trained VLM reasoning backbone; A VL-to-Action module that leverages a set of learnable action queries for parallel action decoding; A refinement module that enhances action prediction performance. As illustrated in Fig. 1, these queries are guided with spatial awareness and undergo self-attention and cross-attention with the VLM simultaneously. By employing additional learnable queries, Reasoning-VLA can predict action chunks in a single step, rather than generating actions token by token, as required in autoregressive approaches. The features from these action queries, together with intermediate VLM representations, are subsequently processed by a series of refinement modules to produce the final action trajectories. The architectural design of Reasoning-VLA offers four key advantages:

1. Leverages the reasoning capabilities of the VLM for more informed and context-aware action generation.

2. Parallel action generation via action queries enables significantly higher inference speed compared to autoregressive.

3. Learnable action queries are guided with implicit spatial enhancement, improving model performance.

4. Refinement modules interact with intermediate hidden states to enhance feature representation and trajectory accuracy.

### 3.3. Learnable Action Queries and Implicit Spatial Guidance

#### 3.3.1. LEARNABLE ACTION QUERIES

We initialize a set of learnable action queries $AQ \in T \times N \times D$. Here $T$ is the number of future time steps to be predicted, $N$ is the dimensions of action trajectory coordinate, $D$ is the feature dimensionality. As shown in Fig.2. Unlike VLMs, which embedded input tokens into embeddings, our action queries are initialized as learnable parameters. This design provides greater flexibility and expressive capacity, enabling parallel prediction of action trajectories, and offering an efficient alternative to sequential token generation.

#### 3.3.2. IMPLICIT SPATIAL GUIDANCE

To enhance accurate spatial awareness and accelerate training convergence, this module initializes learnable action queries with predefined geometric representations derived from ground-truth statistics. These representations guide the learnable action queries to develop robust spatial comprehension capabilities. By enforcing alignment at the initialization stage of action queries, this module enables our VLA to encode richer spatial features, thereby improving action precision. These predefined geometric representations parameters must satisfy two criteria: 1. The predefined parameters must match the shape of action queries; 2. The reasonable initial values that reflect typical action distributions and are equipped with spatial information. Given that the total number of action values is $T \times N$, in our method, we predict future $T$ steps for $N$ coordinates (e.g., $x$, $y$), total action query is $N \times T$. We have to generate $T \times N$ action queries with $D$ dimensions.

Specifically, we extract the action trajectory values of each frame firstly (each frame have $N \times T$ action values, the total action trajectory values are $D_{all} \times N \times T$, where $D_{all}$ is the total number of frames in our datasets), then we calculate the mean action values, such as, $x_1, y_1, x_2, y_2, x_i, y_i, ..., x_N, y_N$, each $x_i, y_i$ represents the average coordinate for the corresponding position. To match the feature dimension of action queries, we extend the $N \times T$ action values to $N \times T \times D$, by sampling $D$ values from a Gaussian distribution with the previously calculated mean and variance. This procedure completes the initialization of the learnable action queries, providing a well-structured and informative starting point for training.

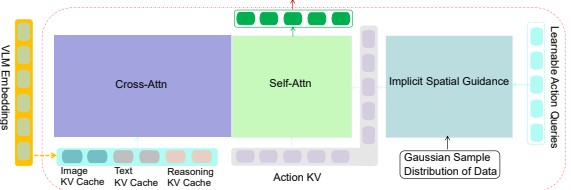

*Figure 2.* The action module interacts with the vision-language model (VLM). The learnable action queries are initialized using a Gaussian distribution derived from the ground-truth (GT) action data. Through self-attention and cross-attention mechanisms with the reasoning VLM, the model transfers the generalized reasoning capability from the VL to A.

### 3.4. How Do Actions Interact with Vision-Language Reasoning?

Unlike autoregression-based or diffusion-based vision-language-action (VLA) methods, our approach employs independent learnable action queries to predict action trajectories. Consequently, the interaction between the action module and the vision-language model (VLM) differs substantially. Since the action queries are not tied to the VLM's token representations, they first perform self-attention and then interact with the VLM through cross-attention, as illustrated in Fig. 2. Through these attention mechanisms, the action queries can extract rich and semantically meaningful information from the VLM's hidden states, which contain extensive reasoning content. This interaction strategy provides a significant advantage: the action queries can generate all expected actions in parallel during a single forward pass, enabling efficient action chunking. This contrasts with autoregression-based VLAs that require sequential token-by-token processing. Our approach reduces action generation from more than $N \times T$ sequential passes to a single pass, substantially improving both training and inference efficiency. Furthermore, we eliminate the discretization process used in autoregressive VLAs, which often degrades fine-grained action details. In addition, we replace the causal attention mask with bidirectional mask, allowing models to predict all actions simultaneously.

### 3.5. Action Refinement Module

To further enhance the representation quality and accuracy of the predicted action trajectories, we introduce an Action Refinement Module (ARM). Specifically, the ARM takes the selected hidden states of the action queries as input and refines them through a combination of multilayer perceptron (MLP) and attention mechanisms. Unlike next-token prediction methods (e.g., $\pi_0$), which employ discrete action representations, our approach adopts a regression-based strategy to generate continuous actions. This design preserves the efficiency benefits of parallel action prediction while improving the precision and smoothness of the resulting action trajectories.

## 3.6. SFT and RL

Drawing inspiration from recent advances in VLMs, we employ two complementary training strategies to enhance the generalization ability of our model: supervised fine-tuning (SFT) and reinforcement learning (RL) fine-tuning.

**SFT.** In this stage, we utilize our unified reasoning dataset to construct structured reasoning chains. Prior studies in VLMs have shown that base models tend to generate tangential or unstructured responses without supervised fine-tuning. Therefore, the SFT process is essential for establishing a solid foundation for subsequent RL training. Reasoning-VLA demonstrates excellent performance on the unified reasoning dataset after SFT.

**RL.** Although SFT effectively fits the training data, it often struggles to generalize to unseen or out-of-distribution scenarios. To address this limitation, we apply the GRPO (Shao et al., 2024) during RL fine-tuning. Unlike conventional policy-based methods, GRPO replaces the critic model—typically as large as the policy model—with an estimation of group scores. This design not only simplifies the overall architecture but also significantly reduces computational overhead during training. The rule-based reward functions used for RL optimization are introduced in the following subsection.

## 3.7. Reward Functions

Normally, AD methods use BEV 2-dimensions coordinates $x, y$ to optimize the loss function, while neglecting physical trajectory constraints and vehicle dynamics. In our RL fine-tuning stage, we design two types of verifiable reward functions to more accurately evaluate and enhance the quality of generated responses.

**Physical Trajectory Reward** Different from most regression-based reward functions that employ the mean squared Euclidean distance $\frac{1}{N}\sum_{i=1}^{N}(x^i - x^i_{\text{gt}})^2$, we adopt a weighted Euclidean distance to better align the predicted coordinates with the ground-truth trajectories. Specifically, our physical trajectory reward is defined as:

$$r_{\text{traj}} = 1 - \frac{1}{N}\sum_{i=1}^{N}\gamma^i\left(\alpha(x^i - x^i_{\text{gt}})^2 + \beta(y^i - y^i_{\text{gt}})^2\right) \quad (1)$$

where $N$ is the number of trajectory steps, $x^i$ and $y^i$ are the predicted coordinates at the $i$-th time step, and $x^i_{\text{gt}}$ and $y^i_{\text{gt}}$ are their corresponding ground-truth values. Because the $x$ and $y$ coordinates in autonomous driving often differ in scale, the weighting factors $\alpha$ and $\beta$ are introduced to balance their respective contributions to the reward. The term $\gamma^i$ represents a discount factor that reduces the influence of future trajectory points. This reward function encourages the autonomous vehicle to follow the desired route by penalizing deviations across the entire trajectory.

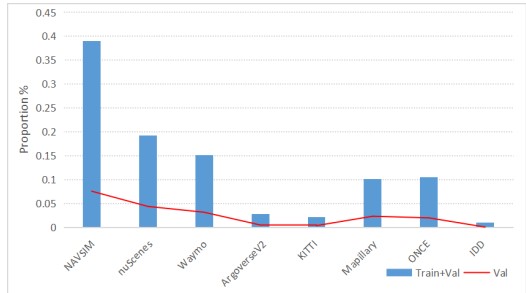

*Figure 3.* Statistical distribution of the unified dataset.

**Vehicle Dynamic Reward** In autonomous driving, most existing studies rarely incorporate vehicle kinematic and dynamic constraints into motion trajectory prediction. However, these constraints exert a non-negligible influence on the vehicle's behavior and overall driving safety. To address this limitation, we propose a Vehicle Dynamics Reward that explicitly accounts for steering and acceleration to constrain the limitations of real-world vehicle dynamics. This design establishes a dynamic constraint optimization objective that ensures physically feasible and stable motion trajectories. The generated action trajectories are governed by both steering kinematics and acceleration dynamics. Specifically, the maximum steering angle is limited to 40 degrees, and the maximum acceleration is constrained to 0.6 gravity. Moreover, abrupt changes in steering or acceleration may lead to vehicle instability or discomfort for passengers. To achieve comfortable and safe driving behavior, the steering constraint reward is defined as:

$$r_{\text{steer}} = \frac{1}{N-1}\sum_{j=1}^{N-1}\begin{cases}1, |\left(y^j - y^{j-1}\right)/\left(x^j - x^{j-1}\right)| < 0.84 \\ 0, |\left(y^j - y^{j-1}\right)/\left(x^j - x^{j-1}\right)| \geq 0.84\end{cases}$$
$$(2)$$

where $(x^j, y^j)$ and $(x^{j-1}, y^{j-1})$ respectively denote the predicted coordinates at $j-th$ and $(j-1)-th$ time step. In this reward function, we reward with 1 when the turning angle is less than 0.84.

We further introduce an Acceleration Reward to constrain non-physical vehicle dynamics. The acceleration reward is defined as:

$$acc_j = \frac{\sqrt{(x^{j+1} - x^j)^2 + (y^{j+1} - y^j)^2}}{T^2}$$
$$- \frac{\sqrt{(x^j - x^{j-1})^2 + (y^j - y^{j-1})^2}}{T^2} \quad (3)$$

$$r_{\text{acc}} = \frac{1}{N-2}\sum_{j=1}^{N-2}\begin{cases}1, & |acc_j| < 6 \\ 0, & |acc_j| \geq 6\end{cases} \quad (4)$$

Here $N$ is the number of trajectory steps, $T$ is the time interval between consecutive actions, $j$ is the $j$-th trajectory step.

*Table 1.* **Open-loop performance on the nuScenes dataset.** Our fully generalized methods, Reasoning-VLA-3B and Reasoning-VLA-7B, follow the complete SFT and RL training process described in the Methods section. The training dataset is our unified dataset, which is constructed from eight public datasets. The validation dataset comprises the corresponding nuScenes validation clips from the unified dataset. Reasoning-VLA-7B+ is fine-tuned with an additional RL process using the corresponding nuScenes training clips from the unified dataset. *: Official checkpoints re-validated with corrected metrics, sourced from (Hu et al., 2022). †: Reproduced with our united dataset(same train and val datasets with Reasoning-VLA. **Reasoning-VLA-7B represents our general model.**

| Methods | L2 (m) ↓ | | | | Collision Rate (%) ↓ | | | |
|---|---|---|---|---|---|---|---|---|
| | 1s | 2s | 3s | Avg. | 1s | 2s | 3s | Avg. |
| **End2End Autonomous Driving Methods** | | | | | | | | |
| ST-P3(Hu et al., 2022) | 1.33 | 2.11 | 2.90 | 2.11 | 0.23 | 0.62 | 1.27 | 0.71 |
| UniAD(Hu et al., 2023)* | 0.45 | 0.70 | 1.04 | 0.73 | 0.62 | 0.58 | 0.63 | 0.61 |
| VAD(Jiang et al., 2023)* | 0.41 | 0.70 | 1.05 | 0.72 | 0.07 | 0.17 | 0.41 | 0.22 |
| PPAD(Chen et al., 2024b) | 0.30 | 0.69 | 1.26 | 0.75 | 0.03 | 0.22 | 0.73 | 0.33 |
| SparseDrive(Sun et al., 2024) | 0.29 | 0.63 | 0.97 | 0.63 | 0.03 | 0.09 | 0.19 | 0.10 |
| **VLM & VLA Autonomous Driving Methods** | | | | | | | | |
| DriveVLM-Dual(Tian et al., 2024) | 0.15 | 0.29 | 0.48 | 0.31 | 0.05 | 0.08 | 0.17 | 0.10 |
| OmniDrive(Wang et al., 2025) | 0.14 | 0.29 | 0.55 | 0.33 | 0.00 | 0.13 | 0.78 | 0.30 |
| EMMA+(Hwang et al., 2025) | 0.13 | 0.27 | 0.48 | 0.29 | – | – | – | – |
| AutoVLA (Zhou et al., 2025) | 0.21 | 0.38 | 0.60 | 0.40 | 0.13 | 0.18 | 0.28 | 0.20 |
| AutoDrive-$R^2$ (Yuan et al., 2025)† | 0.16 | 0.29 | 0.47 | 0.31 | 0.03 | 0.12 | 0.30 | 0.15 |
| Impromptu-VLA(Chi et al., 2025)† | 0.14 | 0.30 | 0.51 | 0.32 | – | – | – | – |
| Impromptu-VLA(Chi et al., 2025) | 0.13 | 0.27 | 0.53 | 0.30 | – | – | – | – |
| **Our Reasoning-VLA Methods** | | | | | | | | |
| Reasoning-VLA-3B | 0.08 | 0.33 | 0.48 | 0.30 | 0.04 | 0.13 | 0.23 | 0.13 |
| **Reasoning-VLA-7B** | 0.05 | 0.20 | 0.44 | 0.23 | 0.01 | 0.07 | 0.15 | 0.08 |
| Reasoning-VLA-7B+ | **0.05** | **0.19** | **0.41** | **0.22** | 0.02 | **0.06** | **0.13** | **0.07** |

As shown in equation above, the reward function effectively constrains steering and acceleration within physical limits, ensuring that the generated action trajectories are both physically realizable and socially acceptable in mixed traffic scenarios. This design further reinforces the autonomous driving system's ability to maintain stable and reasonable motion patterns, which are essential for safe and comfortable driving. The final reward $r_{total}$ is defined as the weighted sum of $r_{traj}$, $r_{steer}$ and $r_{acc}$.

$$r_{\text{total}} = \theta_1 r_{\text{traj}} + \theta_2 r_{\text{steer}} + \theta_3 r_{\text{acc}} \tag{5}$$

Here, $\theta_1, \theta_2, \theta_3$ are coefficients that balance the contributions of each sub-reward.

## 4. Unified Datasets

To capture diverse driving scenarios and further improve generalization, we specifically selected eight widely used autonomous driving datasets as the foundation for our unified dataset: NAVSIM (Dauner et al., 2024), nuScenes (Caesar et al., 2020), Waymo (Sun et al., 2020), Argoverse-V2 (Wilson et al., 2023), KITTI (Geiger et al., 2013), Mapillary (Neuhold et al., 2017), ONCE (Mao et al., 2021), and IDD (Varma et al., 2019). However, many of the original clips lack meaningful text-image associations and often have coarse annotations, limiting their suitability for vision-language-action (VLA) reasoning and creative generation.

From these sources, we carefully selected over 75,000 high-

quality clips to form a reasoning-intensive dataset. Each clip was processed using a strong reasoning VLM to generate Chain-of-Thought descriptions, followed by comprehensive human verification and visualization to ensure correctness and annotation quality. The final dataset is provided in a consistent, standardized format, facilitating downstream training and evaluation. The statistical analysis of the resulting unified dataset is presented in Fig. 3. The processing pipeline for the unified dataset is illustrated in Fig. 5 of Appendix B. A representation of the dataset is also provided in the Appendix B.

## 5. Experiments

We conduct experiments to evaluate Reasoning-VLA as an efficient VLA method for autonomous driving, assess the effectiveness of our training process, and explore its potential as a unified base model for specific autonomous driving tasks. The experiments are designed to answer the following questions:

1. How does Reasoning-VLA compare to prior autonomous driving VLA, when evaluated across multiple datasets and under various generalization scenarios?

2. How does each design affect the performance of fine-tuned Reasoning-VLA on general autonomous driving tasks?

3. Can the design of Reasoning-VLA influence inference efficiency (action generation throughput and latency) and

*Table 2.* **Closed-loop performance on the NeuroNCAP.** We utilize the challenging closed-loop NeuroNCAP simulator to emulate a wide range of complex real-world driving scenarios. Since NeuroNCAP offers a standardized benchmark and evaluation metrics commonly used by other methods, we adhered to its recommended configuration. The Reasoning-VLA modules and fine-tuning process are identical to those employed in the open-loop evaluation. *: Sourced from (Ljungbergh & et al., 2024; Zhang et al., 2025a). †: Reproduced with our united dataset(same train and val datasets with Reasoning-VLA.

| Methods | NeuroNCAP Score ↑ | | | | Collision Rate (%) ↓ | | | |
|---|---|---|---|---|---|---|---|---|
| | Stationary | Frontal | Side | Avg. | Stationary | Frontal | Side | Avg. |
| **End2End & VLA Autonomous Driving Methods** | | | | | | | | |
| UniAD(Hu et al., 2023)* | 0.84 | 0.10 | 1.26 | 0.73 | 87.8 | 98.4 | 79.6 | 88.6 |
| VAD(Jiang et al., 2023)* | 0.47 | 0.04 | 1.45 | 0.66 | 96.2 | 99.6 | 81.6 | 92.5 |
| SparseDrive(Sun et al., 2024)* | – | – | – | 0.92 | – | – | – | 93.9 |
| BridgeAD-B(Zhang et al., 2025a)* | – | – | – | 1.60 | – | – | – | 72.6 |
| AutoDrive-$R^2$ (Yuan et al., 2025)† | 1.64 | 2.11 | 1.93 | 1.89 | 68.2 | 59.5 | 62.2 | 63.3 |
| Impromptu-VLA(Chi et al., 2025)† | 1.50 | 2.08 | 1.96 | 1.85 | 71.3 | 58.1 | 69.5 | 66.3 |
| Impromptu-VLA(Chi et al., 2025) | 1.77 | 2.31 | 2.10 | 2.15 | 70.0 | 59.0 | 65.0 | 65.5 |
| **Our Reasoning-VLA Methods** | | | | | | | | |
| Reasoning-VLA-3B | 1.88 | 2.29 | 1.94 | 2.04 | 63.7 | 60.4 | 64.1 | 62.7 |
| **Reasoning-VLA-7B** | 1.93 | **2.57** | **2.24** | **2.25** | 59.8 | **56.0** | **62.2** | **59.4** |
| Reasoning-VLA-7B+ | **2.06** | 2.33 | 2.17 | 2.19 | **57.9** | 57.4 | 64.0 | 59.8 |

make it more accessible?

### 5.1. Experiment Setups

In our experiments, we mainly evaluate Reasoning-VLA's performance on unified AD datasets, which are constructed from eight autonomous driving datasets. To fairly compare with existing methods, we retain the original training and testing splits of each dataset. During training, we shuffle the unified datasets and fine-tune Reasoning-VLA sequentially using SFT followed by RL. The decay learning rate are start from 5e-4 and e-6 form SFT and RL separately, the accumulated size is 2. Training is performed for 4 epochs for SFT and 1 epoch for RL, using a total batch size of 8 distributed across 8 H200 GPUs. For open-loop evaluation, we use the same testing and validation clips as employed by prior methods. For closed-loop evaluation, the model is tested on the NeuroNCAP benchmark to enable a fair comparison with other approaches.

### 5.2. Main Comparison Results

#### 5.2.1. OPEN-LOOP EVALUATION

Since our goal is to propose a generalized VLA model for autonomous driving, we train our model using the proposed unified dataset. To ensure a fair comparison with prior methods, we adopt the same validation splits and report results on open-loop benchmarks. The open-loop performance on the nuScenes dataset is summarized in Table 1. Three main models are presented in this table: **Reasoning-VLA-3B:** Based on Qwen2.5-VL-3B, trained using the complete SFT and RL process. **Reasoning-VLA-7B:** Based on Qwen2.5-VL-7B and fine-tuned using the SFT and RL process. **Reasoning-VLA-7B+:** Similar to Reasoning-VLA-7B, but additionally fine-tuned with RL on selected nuScenes training clips from the unified dataset. Our results show that the purely general-

ized model, **Reasoning-VLA-7B**, surpasses previous works across benchmarks, achieving substantial improvements of +23.3% in average L2 and +20.0% in average Collision Rate over the existing best methods. Reasoning-VLA-3B also achieves results comparable to state-of-the-art methods. When fine-tuned with GRPO on specific datasets (i.e., selected nuScenes training clips from the unified dataset), our generalized model demonstrates excellent task-specific performance. As shown in the last row of Table 1, the additional fine-tuning further improves performance across all time intervals: Reasoning-VLA-7B+ achieves increases of 4.3% and 12.5% over Reasoning-VLA-7B in average L2 and Collision Rate, respectively. These results indicate that our approach provides significant improvements in open-loop evaluation, highlighting the strong generalization capability of the Reasoning-VLA architecture. Consequently, it can serve as an effective base model for downstream autonomous driving tasks.

#### 5.2.2. CLOSED-LOOP EVALUATION

We use NeuroNCAP (Ljungbergh et al., 2024) as the closed-loop real-world simulator because it provides renderings of novel, unseen scenarios. As shown in Table 2, the three main models evaluated are the same as those in the open-loop experiments. Our methods demonstrate significant advantages in closed-loop performance on NeuroNCAP. The generalized model, Reasoning-VLA-7B, substantially outperforms prior methods in terms of NeuroNCAP Score and Collision Rate, achieving an average NeuroNCAP Score of 2.25 and an average Collision Rate of 59.4. When additionally fine-tuned with RL on selected nuScenes training clips from the unified dataset, performance on stationary scenarios shows slight improvement; however, overall performance decreases. This is because the smaller nuScenes dataset adjusts the model to fit that specific data, thereby reducing

*Table 3.* **Generalized performance on our unifed dataset.** We trained two models using the unified dataset: Reasoning-VLA-7B + SFT: This model is fine-tuned using only supervised fine-tuning (SFT). Reasoning-VLA-7B + SFT + RL: This model undergoes the full training process, including both SFT and reinforcement learning (RL). The training dataset for both models is the unified training dataset. For evaluation, the dataset splits follow the recommendations provided by each original dataset.

| Datasets | Reasoning-VLA-7B + SFT | | | | Reasoning-VLA-7B + SFT + RL | | | |
|---|---|---|---|---|---|---|---|---|
| | L2 (m) $\downarrow$ | | | | L2 (m) $\downarrow$ | | | |
| | 1s | 2s | 3s | Avg. | 1s | 2s | 3s | Avg. |
| NAVSIM(Dauner et al., 2024) | 0.05 | 0.18 | 0.43 | 0.22 | 0.04 | 0.18 | 0.41 | 0.21 |
| nuScenes(Caesar et al., 2020) | 0.06 | 0.23 | 0.48 | 0.26 | 0.05 | 0.20 | 0.44 | 0.23 |
| Waymo(Sun et al., 2020) | 0.04 | 0.15 | 0.44 | 0.21 | 0.03 | 0.14 | 0.48 | 0.22 |
| Argoverse-V2(Wilson et al., 2023) | 0.01 | 0.13 | 0.45 | 0.20 | 0.01 | 0.14 | 0.43 | 0.19 |
| KITTI(Geiger et al., 2013) | 0.02 | 0.15 | 0.48 | 0.22 | 0.01 | 0.15 | 0.43 | 0.20 |
| Mapillary(Neuhold et al., 2017) | 0.04 | 0.44 | 0.92 | 0.47 | 0.04 | 0.41 | 1.01 | 0.49 |
| ONCE(Mao et al., 2021) | 0.07 | 0.49 | 0.87 | 0.48 | 0.06 | 0.43 | 0.90 | 0.46 |
| IDD(Varma et al., 2019) | 0.02 | 0.29 | 0.77 | 0.36 | 0.03 | 0.27 | 0.81 | 0.37 |
| Unified | 0.05 | 0.20 | 0.47 | 0.24 | 0.05 | 0.20 | 0.44 | 0.23 |

its generalization ability in closed-loop evaluation. Notably, even Reasoning-VLA-3B surpasses competing methods, achieving more than a 4.3% improvement in average Collision Rate. These results demonstrate the strong generalization capability of our model, particularly in closed-loop environments that involve previously unseen scenarios.

*Table 4.* **Ablation study of components contributions.** R-VLA (Reasoning-VLA) is a 7B-parameter model. All experiments were conducted on our unified dataset and evaluated using a selected subset of the nuScenes dataset extracted from the unified dataset.

| Methods | L2 (m) $\downarrow$ | | | |
|---|---|---|---|---|
| | 1s | 2s | 3s | Avg. |
| Qwen2.5-VL-7B | 0.46 | 1.33 | 2.55 | 1.45 |
| R-VLA(w/o AQ)+SFT | 0.09 | 0.31 | 0.55 | 0.32 |
| R-VLA(w/o AQ)+SFT+RL | 0.08 | 0.30 | 0.52 | 0.30 |
| R-VLA(w/o AQ-Init)+SFT | 0.06 | 0.27 | 0.55 | 0.29 |
| R-VLA(w/o AQ-Init)+SFT+RL | 0.08 | 0.23 | 0.50 | 0.27 |
| R-VLA(w/o ARM)+SFT | 0.06 | 0.28 | 0.53 | 0.29 |
| R-VLA(w/o ARM)+SFT+RL | 0.05 | 0.24 | 0.57 | 0.29 |
| R-VLA+SFT | 0.06 | 0.23 | 0.48 | 0.26 |
| R-VLA+SFT+RL | 0.05 | 0.20 | 0.44 | 0.23 |

### 5.2.3. GENERALIZED PERFORMANCE

To evaluate the generalization capability of Reasoning-VLA, we tested Reasoning-VLA-7B on eight sub-datasets. As shown in Table 3, our results demonstrate that Reasoning-VLA exhibits strong generalization performance. The model was trained on the unified dataset and evaluated separately on each sub-dataset. We observed that the L2 performance for each sub-dataset closely matches that of the overall unified validation dataset. The variance of the L2 values is minimal, with variance values of 0.012 and 0.014 for average L2 in Reasoning-VLA-7B with SFT and Reasoning-VLA-7B with SFT plus RL, respectively. These results indicate that our method maintains robust generalization across different driving scenarios and vehicle configurations.

### 5.3. Ablation Study

#### 5.3.1. KEY DESIGN CONTRIBUTIONS

We conducted ablation studies to evaluate the effectiveness of key component designs, with results summarized in Table 4. Five experimental groups were constructed using different combinations of model components. As shown in Group-1 of Tab.4, the evaluation result of original Qwen2.5-VL-7B in poor performance. Differently, in Group-2, we replaced the learnable action queries with non-learnable queries and trained the model exclusively on the unified dataset. This modification yielded suboptimal results, achieving an average L2 of 0.32 and 0.30 with SFT and SFT+RL fine-tuning, respectively. In Group-3, only the implicit spatial enhance strategy of learnable action query was removed, resulting in a slight performance degradation compared to the full Reasoning-VLA model (Group-5), which proves the effectiveness of spatial enhancement. The results from Groups 2 and 3 suggest that learnable action queries significantly contribute to the model's ability to generalize across diverse autonomous driving scenarios, thereby enhancing the overall performance of Reasoning-VLA. In Group-4, the action refinement module was removed, and actions were directly regressed from parallel action outputs using an MLP. This strategy led to a modest performance drop relative to Group-5. Overall, these ablation studies demonstrate that each component of Reasoning-VLA contributes to its final performance, confirming the effectiveness of the proposed design.

### 5.4. Zero-Shot Performance

We also conducted a "zero shot" experiment to further validate the generalization capability of our model. Specifically, the unified dataset was partitioned into distinct scenarios, where the NAVSIM, Waymo, KITTI, and ONCE sub-datasets were used for training, and the remaining four sub-datasets served as validation sets. As shown in Table

*Table 5.* **Zero shot performance on our unified dataset.** The unified dataset is divided into two parts: the training set, which includes data from NAVSIM, Waymo, KITTI, and ONCE, and the evaluation sets, which consist of the remaining four sub-datasets along with the unified validation set.

| Datasets | Reasoning-VLA-7B + SFT | | | | Reasoning-VLA-7B + SFT + RL | | | |
|---|---|---|---|---|---|---|---|---|
| | L2 (m) ↓ | | | | L2 (m) ↓ | | | |
| | 1s | 2s | 3s | Avg. | 1s | 2s | 3s | Avg. |
| nuScenes(Caesar et al., 2020) | 0.07 | 0.24 | 0.52 | 0.28 | 0.08 | 0.26 | 0.50 | 0.28 |
| Argoverse-V2(Wilson et al., 2023) | 0.03 | 0.18 | 0.59 | 0.27 | 0.04 | 0.18 | 0.55 | 0.26 |
| Mapillary(Neuhold et al., 2017) | 0.08 | 0.59 | 1.09 | 0.59 | 0.07 | 0.57 | 1.01 | 0.55 |
| IDD(Varma et al., 2019) | 0.09 | 0.41 | 0.96 | 0.49 | 0.08 | 0.38 | 0.93 | 0.46 |
| All | 0.07 | 0.28 | 0.57 | 0.31 | 0.07 | 0.27 | 0.53 | 0.29 |

*Table 6.* **Performance on the NAVSIM.** *: Sourced from (Dauner et al., 2024).

| Methods | NC↑ | DAC↑ | TTC↑ | Comfort↑ | EP↑ | PDMS↑ |
|---|---|---|---|---|---|---|
| TransFuser(Chitta et al., 2022)* | 97.7 | 92.8 | 92.8 | 100 | 79.2 | 84.0 |
| UniAD(Hu et al., 2023)* | 97.8 | 91.9 | 92.9 | 100 | 78.8 | 83.4 |
| Para-Drive(Weng et al., 2024)* | 97.9 | 92.4 | 93.0 | 99.8 | 79.3 | 84.0 |
| Reasoning-VLA-7B | 97.8 | **93.2** | **98.1** | 99.8 | **80.7** | **91.7** |

5, our method exhibits strong generalization performance on unseen datasets. This experiment confirms that the proposed Reasoning-VLA possesses robust adaptability to new driving scenarios and tasks, highlighting its potential as a general-purpose autonomous driving framework.

### 5.5. Performance on NAVSIM

Moreover, Tab.6 demonstrates the evaluation results on the NAVSIM evaluation. Compared to the Para-Drive method, our approach achieves respective improvements of 0.8, 5.1, 1.4, and 7.7 in DAC, TTC, EP, and PDMS metrics. Overall, the proposed model consistently delivers accurate and reliable predictions across the NAVSIM evaluations, establishing its state-of-the-art performance and strong generalization capability.

## 6. Conclusions

This paper presents a general and efficient VLA framework based on a reasoning-enhanced vision-language model for autonomous driving. The proposed method introduces learnable action queries, initialized through Gaussian sampling from ground-truth trajectories, which interact with reasoning-augmented vision–language features to generate continuous action trajectories in parallel, thereby significantly improving inference efficiency. To enhance generalization, we unify eight existing autonomous driving datasets into a standardized, reasoning-based, and easy-to-use unified dataset. Following supervised fine-tuning (SFT) and reinforcement learning (RL) optimization, our method demonstrates outstanding performance and strong generalization capabilities in autonomous driving tasks.

## Acknowledgements

This research is supported by the National Research Foundation, Singapore under its National Large Language Models Funding Initiative (AISG Award No: AISG-NMLP-2024-002). Any opinions, findings and conclusions or recommendations expressed in this material are those of the author(s) and do not reflect the views of National Research Foundation, Singapore.

## Impact Statement

This paper presents work whose goal is to advance the field of Machine Learning. There are many potential societal consequences of our work, none which we feel must be specifically highlighted here.

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

# A. Appendix A

## A.1. More Ablation Studies

### A.1.1. THE SOURCE OF PERFORMANCE: GENERALIZATION ABILITY OR DATA CONTRIBUTION?

To further demonstrate that the SOTA performance of Reasoning-VLA arises from its generalization capabilities rather than from reliance on a specific dataset, we conducted two types of experiments, as shown in Tables 7 and 8. We evaluated two types of fine-tuned models:

**Reasoning-VLA Fine-tuned on the nuScenes Dataset:** The Reasoning-VLA (3B and 7B) models were fine-tuned exclusively on the selected nuScenes subset extracted from the unified dataset.

**Reasoning-VLA Fine-tuned on the Unified Dataset:** The Reasoning-VLA (3B and 7B) models were fine-tuned on the entire unified dataset.

Open-loop evaluations were performed on the corresponding nuScenes validation subset of the unified dataset. As shown in Table 7, the Reasoning-VLA models fine-tuned on the unified dataset outperform those fine-tuned solely on the nuScenes subset in terms of average L2 error and collision rate. Specifically, Reasoning-VLA-7B fine-tuned on the selected nuScenes subset achieves an average L2 error of 0.25 and a collision rate of 0.10, which are 8.7% and 25% lower, respectively, than the general Reasoning-VLA-7B fine-tuned on the unified dataset. Closed-loop evaluations, summarized in Table 8, further indicate that models fine-tuned on the unified dataset outperform those trained only on the nuScenes subset across all metrics. These results confirm that Reasoning-VLA possesses strong generalization capabilities in autonomous driving scenarios, comparable to those observed in VLMs.

*Table 7.* **Generalization performance on the Open-loop Metrics.**

| Methods | L2 (m) ↓ | | | | Collision Rate (%) ↓ | | | |
|---|---|---|---|---|---|---|---|---|
| | 1s | 2s | 3s | Avg. | 1s | 2s | 3s | Avg. |
| **Reasoning-VLA Finetuned with nuScenes Dataset** | | | | | | | | |
| Reasoning-VLA-3B | 0.10 | 0.38 | 0.51 | 0.33 | 0.05 | 0.13 | 0.27 | 0.15 |
| Reasoning-VLA-7B | 0.07 | 0.23 | 0.46 | 0.25 | 0.01 | 0.08 | 0.20 | 0.10 |
| **Reasoning-VLA Finetuned with Our Unified Dataset** | | | | | | | | |
| Reasoning-VLA-3B | 0.08 | 0.33 | 0.48 | 0.30 | 0.04 | 0.13 | 0.23 | 0.13 |
| Reasoning-VLA-7B | 0.05 | 0.20 | 0.44 | 0.23 | 0.01 | 0.07 | 0.15 | 0.08 |

*Table 8.* **Generalization performance on the Closed-loop Metrics.**

| Methods | NeuroNCAP Score ↑ | | | | Collision Rate (%) ↓ | | | |
|---|---|---|---|---|---|---|---|---|
| | Stationary | Frontal | Side | Avg. | Stationary | Frontal | Side | Avg. |
| **Reasoning-VLA Finetuned with nuScenes Dataset** | | | | | | | | |
| Reasoning-VLA-3B | 1.67 | 2.16 | 1.83 | 1.89 | 69.0 | 63.3 | 66.6 | 66.3 |
| Reasoning-VLA-7B | 1.79 | 2.44 | 2.12 | 2.12 | 61.5 | 57.1 | 65.1 | 61.3 |
| **Reasoning-VLA Finetuned with Our Unified Dataset** | | | | | | | | |
| Reasoning-VLA-3B | 1.88 | 2.29 | 1.94 | 2.04 | 63.7 | 60.4 | 64.1 | 62.7 |
| Reasoning-VLA-7B | 1.93 | 2.57 | 2.24 | 2.25 | 59.8 | 56.0 | 62.2 | 59.4 |

### A.1.2. INFERENCE EFFICIENCY

To evaluate the inference efficiency of Reasoning-VLA, we conducted experiments summarized in Table 9. Compared to existing autoregression-based VLMs, our method achieves superior performance using the same backbone. Reasoning-VLA can generate multiple future trajectories (e.g., 6 or 10 trajectories) in a single inference step, whereas autoregression-based VLA/VLMs must generate these trajectories sequentially. Even when employing the efficient bin-tokenizer proposed by OpenVLA (Kim et al., 2024) and $\pi_0$ (Black et al., 2024), these methods require at least 12 to 20 steps to generate the

desired trajectories, including both reasoning and trajectory tokens. Our experiments show that Reasoning-VLA achieves a generation speed of 0.089s per inference for 10 trajectories using vLLM, which is approximately 61 times faster than the autoregression-based Qwen2.5-VL-7B for the same number of trajectories. These results clearly demonstrate the superior inference efficiency of the Reasoning-VLA design.

*Table 9.* **The Efficiency Comparisons.** Steps: Theoretical number of VLM inference steps required to complete a single prediction process. Speed(s): Measured inference time to generate a complete prediction process. All experiments were conducted on an NVIDIA H200 GPU using vLLM. Traj: Number of predicted trajectories.

| Methods | Steps | Speed(s) |
|---|---|---|
| Qwen2.5-VL-7B(6 Traj) | $\gg 12$ | 5.396 |
| Qwen2.5-VL-7B(10 Traj) | $\gg 20$ | 5.472 |
| Reasoning-VLA-7B(6 Traj) | 1 | 0.081 |
| Reasoning-VLA-7B(10 Traj) | 1 | 0.089 |

### A.1.3. MODEL SIZE

As is well known, the performance of LLMs and VLMs generally improves with an increase in model parameters. To analyze the impact of model size, we compare the 3B and 7B variants of our Reasoning-VLA. As shown in Tables 1 and 2, the Reasoning-VLA-7B model achieves superior performance, with an average L2 error of 0.23 and an average NeuroNCAP Score of 2.25, representing improvements of 30.4% and 9.3%, respectively, over the Reasoning-VLA-3B model. This performance gap indicates that larger models inherently possess stronger representational capacity, enabling them to capture more complex patterns and achieve better results.

### A.1.4. REASONING ABLATION

To examine the effectiveness of the reasoning component in our method, we conducted an ablation study. As shown in 10, removing the reasoning process from our Reasoning-VLA model results in a 26% performance decrease compared to Reasoning-VLA-7B. These results demonstrate the critical role of the reasoning process in our approach.

*Table 10.* **Ablation study of reasoning contributions.**

| Methods | L2 (m) $\downarrow$ | | | |
|---|---|---|---|---|
| | 1s | 2s | 3s | Avg. |
| Reasoning-VLA-7B w/o Reasoning | 0.08 | 0.26 | 0.53 | 0.29 |
| Reasoning-VLA-7B | 0.05 | 0.20 | 0.44 | 0.23 |

## A.2. Qualitative Results of Action Trajectories

We also provide qualitative results to further demonstrate the effectiveness of Reasoning-VLA. As illustrated in Fig.4, the visualization of predicted action trajectories across eight datasets highlights the strong generalization capability of our method. Notably, Reasoning-VLA produces consistent and accurate trajectory predictions even in previously unseen scenarios, confirming its robustness and adaptability.

## B. Appendix B

### B.1. Unified Dataset

Existing individual autonomous driving datasets are often limited in scope, providing narrow coverage of the diverse scenarios encountered in real-world driving. To address this, we aggregate eight public datasets to construct a unified, reasoning-intensive dataset designed to support Chain-of-Thought generation with a strong reasoning VLM. This unified dataset is organized into a coherent, easy-to-use format to facilitate model training and enhance the generalization capability of Reasoning-VLA.

The processing pipeline for the unified dataset is illustrated in Fig. 5. First, all source datasets are converted into a standardized data format. The resulting image-text pairs are then input into a VLM, which generates detailed reasoning content according to a predefined labeling protocol. This reasoning output undergoes a rule-based verification process,

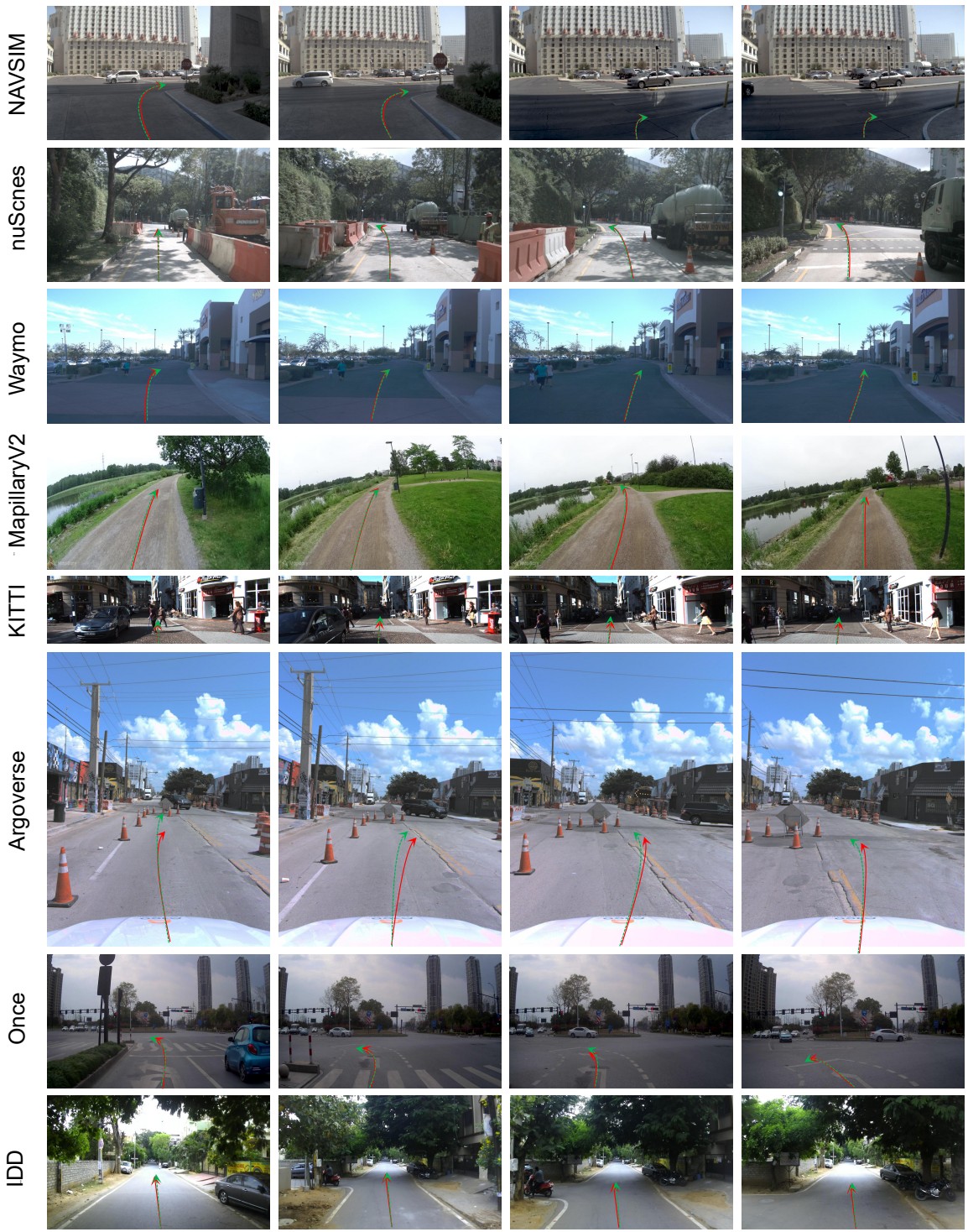

*Figure 4.* **Qualitative Results of Action Trajectories.** Reasoning-VLA predictions on eight different datasets.Red lines denote GT trajectories while green lines represent predicted trajectories.

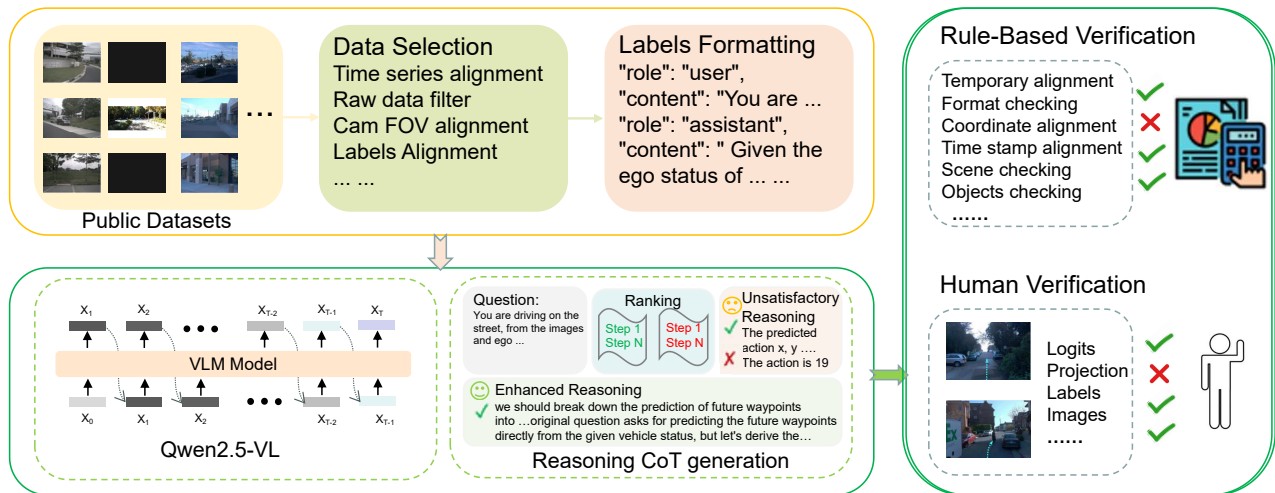

*Figure 5.* Pipeline for generating the unified reasoning dataset.

followed by human review. During human verification, annotators assess the clips along with their associated labels to select the final set of high-quality data.

## B.2. CoT Reasoning Unifided Dataset Format and Prompt

During the SFT stage of our method, we designed a structured input prompt to facilitate the generation of high-quality chain-of-thought (CoT) reasoning data. The prompt template is presented as follows:

```
{'role': 'system', 'content': [{'type': 'text', 'text': 'You are a helpful assistant'}]},

{'role': 'user', 'content': [{'type': 'image', 'image': 'nuScences_441_4000.CAM_FRONT.png
    '}, {'type': 'image', 'image': 'nuScences_441_4000.CAM_LEFT.png'}, {'type': 'image',
    'image': 'nuScences_441_4000.CAM_RIGHT.png'}, {'type': 'text', 'text': "You are an
    autonomous driving agent. You have access to multi-view camera images of a vehicle:
    (1) front view (which you should focus on with the most attention) <image>, (2) front
     right view <image>, and (3) front left view <image>. Your task is to do your best to
     predict future waypoints for the vehicle over the next 10 timesteps, given the
    vehicle's intent inferred from the images. Provided are the previous ego vehicle
    status recorded over the last 3.0 seconds (at 0.5-second intervals). This includes
    the x and y coordinates of the ego vehicle. Positive x means forward direction while
    positive y means leftwards. The data is presented in the format [x, y]:(t-3.0s)
    [-21.95, -0.11], Acceleration: X 0.22, Y 0.21 m/s^2, Velocity: X 6.93, Y 0.0 m/s, (t
    -2.5s) [-18.42, -0.07], Acceleration: X 0.19, Y 0.22 m/s^2, Velocity: X 7.03, Y 0.0 m
    /s, (t-2.0s) [-14.88, -0.05], Acceleration: X 0.26, Y 0.15 m/s^2, Velocity: X 7.16, Y
     0.0 m/s, (t-1.5s) [-11.22, -0.02], Acceleration: X 0.16, Y 0.15 m/s^2, Velocity: X
    7.25, Y 0.0 m/s, (t-1.0s) [-7.15, 0.02], Acceleration: X -0.21, Y 0.16 m/s^2,
    Velocity: X 7.23, Y 0.0 m/s, (t-0.5s) [-3.52, 0.02], Acceleration: X -0.39, Y 0.19 m/
    s^2, Velocity: X 7.09, Y 0.0 m/s, (t-0.0s) [0.0, 0.0], Acceleration: X -0.51, Y -0.06
     m/s^2, Velocity: X 6.88, Y 0.0 m/s\n\nPlease think deeply. Engage in an internal
    dialogue other natural language thought expressions It's a reasoning process. Provide
     your reasoning between the <think> </think> tags, and then give your answer between
    the <answer> </answer> tags. Predicted future movement details for the next 5 seconds
     (sampled at 0.5-second intervals), including BEV location in x and y directions (in
    meters). Positive x means forward direction while positive y means leftwards. The
    output is formatted as [x, y]."}]},

{'role': 'assistant', 'content': [{'type': 'text', 'text': "<think>\nLet me think. To
    rephrase the question in a way that requires Chain-of-Thought reasoning with
    numerical or mathematical expressions, we should break down the prediction of future
    waypoints into smaller steps, starting from understanding the provided data and
```

```
    applying relevant physics equations. \n\nThe original question asks for predicting
    the future waypoints directly from the given vehicle status, but let's derive the
    waypoints through intermediate calculations. \n \nOh, I see. The question now needs
    to be framed in such a way that the responder understands they need.\n</think>\n<
    answer><|place_holder|><|place_holder|><|place_holder|><|place_holder|><|place_holder
    |><|place_holder|><|place_holder|><|place_holder|><|place_holder|><|place_holder|><|
    place_holder|><|place_holder|><|place_holder|><|place_holder|><|place_holder|><|
    place_holder|><|place_holder|><|place_holder|><|place_holder|><|place_holder|></
    answer>"}]},
  {'actions': array([[0.        , 0.        ],
      [0.40046561, 0.39716284],
      [0.39221381, 0.33593741],
      [0.39243497, 0.31284149],
      [0.38875668, 0.28942805],
      [0.38467048, 0.27311695],
      [0.38409407, 0.26829267],
      [0.38829151, 0.27437078],
      [0.3931127 , 0.28797924],
      [0.39968362, 0.29992925]])}
```

## B.3. Training Details

The training details of SFT and RL are illustrated below.

### B.3.1. SFT

```
batch_size 8
gradient_accumulation_steps 2
learning_rate 5e-5
bf16
gradient_checkpointing true
attn_implementation flash_attention_2
num_train_epochs 4
max_grad_norm 5

zero2 config:
{
    "fp16": {
        "enabled": "auto",
        "loss_scale": 0,
        "loss_scale_window": 1000,
        "initial_scale_power": 16,
        "hysteresis": 2,
        "min_loss_scale": 1
    },
    "bf16": {
        "enabled": "auto"
    },
    "optimizer": {
        "type": "AdamW",
        "params": {
            "lr": "auto",
            "betas": "auto",
            "eps": "auto",
            "weight_decay": "auto"
        }
    },
    "zero_optimization": {
        "stage": 2,
        "offload_optimizer": {
            "device": "none",
            "pin_memory": true
        },
```

```
            "allgather_partitions": true,
            "allgather_bucket_size": 2e8,
            "overlap_comm": false,
            "reduce_scatter": true,
            "reduce_bucket_size": 2e8,
            "contiguous_gradients": true
    },
    "gradient_accumulation_steps": "auto",
    "gradient_clipping": "auto",
    "steps_per_print": 100,
    "train_batch_size": "auto",
    "train_micro_batch_size_per_gpu": "auto",
    "wall_clock_breakdown": false
}
```

## B.3.2. RL

```
max_prompt_length 16384
max_completion_length 768
batch_size 8
gradient_accumulation_steps 2
learning_rate 1e-6
lr_scheduler_type "cosine"
weight_decay 0.01
bf16
gradient_checkpointing true
temporal true
len_control true
attn_implementation flash_attention_2
max_pixels 401408
num_train_epochs 1
beta 0.04
max_grad_norm
num_generations 8

zero3 config:

    "fp16": {
        "enabled": "auto",
        "loss_scale": 0,
        "loss_scale_window": 1000,
        "initial_scale_power": 16,
        "hysteresis": 2,
        "min_loss_scale": 1
    },
    "bf16": {
        "enabled": "auto"
    },

    "zero_optimization": {
        "stage": 3,
        "offload_optimizer": {
            "device": "none",
            "pin_memory": true
        },
        "offload_param": {
            "device": "none",
            "pin_memory": true
        },
        "overlap_comm": true,
        "contiguous_gradients": true,
        "sub_group_size": 1e9,
        "reduce_bucket_size": "auto",
```

```
        "stage3_prefetch_bucket_size": "auto",
        "stage3_param_persistence_threshold": "auto",
        "stage3_max_live_parameters": 1e9,
        "stage3_max_reuse_distance": 1e9,
        "stage3_gather_16bit_weights
          _on_model_save": true
    },

    "gradient_accumulation_steps": "auto",
    "gradient_clipping": "auto",
    "steps_per_print": 100,
    "train_batch_size": "auto",
    "train_micro_batch_size_per_gpu": "auto",
    "wall_clock_breakdown": false
}
```

## B.4. Reward Function Implements

Our reward functions significantly influence the RL process. For trajectory reward function:

$$r_{\text{traj}} = 1 - \frac{1}{N} \sum_{i=1}^{N} \gamma^i \left( \alpha(x^i - x_{\text{gt}}^i)^2 + \beta(y^i - y_{\text{gt}}^i)^2 \right) \tag{6}$$

normally, we can select hyper-parameters (1 in this function) to achieve an accurate reward. Some times we need change the 1 to large numbers such as 2. We can also replaced with another easy way:

$$r_{\text{traj}} = 1 - min(1.0, \frac{1}{N} \sum_{i=1}^{N} \gamma^i \left( \alpha(x^i - x_{\text{gt}}^i)^2 + \beta(y^i - y_{\text{gt}}^i)^2 \right)) \tag{7}$$

## B.5. Related Works: Classic Autonomous Driving

Classic autonomous driving (AD) methods have been developed over many years, evolving from modular systems to modern end-to-end learning frameworks (Philion & Fidler, 2020; Huang & Huang, 2022; Zhang et al., 2024; Yuan et al., 2023; Liang et al., 2020; Prakash et al., 2021; Zhang et al., 2023). Early AD systems were typically constructed by cascading these single-task modules into a sequential pipeline (Li et al., 2022b; Wang et al., 2021; Liu et al., 2023; Zhang et al., 2024; Liu et al., 2021). However, such designs suffer from error accumulation, where inaccuracies in upstream tasks propagate through subsequent modules, ultimately degrading overall system performance. To address this issue, recent research has shifted toward end-to-end learning-based approaches that integrate all sub-tasks into a unified framework. Modern open-source end-to-end AD methods increasingly rely on bird's-eye view (BEV) feature representations and generate planning trajectories through cross-interactions among internal components (Hu et al., 2023; Jiang et al., 2023; Chen et al., 2024a; Jia et al., 2023; 2024). Meanwhile, other approaches exploit sparse feature extraction from the 3D environment to directly infer results from image features, thereby avoiding the computational cost of constructing explicit BEV features (Wang et al., 2021; Sun et al., 2024). Collectively, these advances have simplified the traditional multi-stage AD pipeline, marking the beginning of a new era of data-driven autonomous driving.

