# OpenReview forum: "Reasoning-VLA: An Efficient and Spatial-Guided General Vision-Language-Action Reasoning Model for Autonomous Driving"
_ICML.cc/2026/Conference — ICML 2026 regular_

### Official Review · Reviewer_jNHv · 2026-02-15

**Soundness:** 2
**Presentation:** 2
**Significance:** 3
**Originality:** 1
**Overall Recommendation:** 4
**Confidence:** 4

**Summary:**

This paper proposes Reasoning-VLA, which improves upon existing VLM-based frameworks by optimising the output mechanism to accelerate inference speed. Additionally, the authors integrate eight mainstream autonomous driving datasets to construct a unified dataset, including Chain-of-Thought annotations generated by a VLM to facilitate effective VLA training.

**Compliance With Llm Reviewing Policy:**

Affirmed.

**Final Justification:**

The authors have satisfactorily addressed all of my concerns in the rebuttal and accepted the suggested modifications. The additional experiments they provided can demonstrate the efficiency of their proposed method. Without the fourth contribution regarding the RL reward function, the paper still presents some workload and original contributions. Therefore, I am updating my score and recommend a Weak Accept.

**Key Questions For Authors:**

1. Why is there no comparison provided regarding the inference speed of existing VLA methods?
2. Will the dataset mentioned in the paper be made publicly available?

**Limitations:**

No. The paper lacks a dedicated discussion on the limitations of the proposed method. I suggest the authors add a paragraph (e.g., in the Conclusion or a separate Discussion section) addressing:

- Ambiguity in Trajectory Regression: The model lacks high-level navigation commands (e.g., "turn left"). The authors should discuss the limitations of using a regression-based loss in ambiguous scenarios (e.g., intersections) where multiple valid futures exist, which potentially leads to mode averaging.

- Dependence on Teacher Model Quality: Since the training relies on CoT data generated by a larger VLM, the model is susceptible to learning hallucinations or biases present in the teacher model. This dependency should be acknowledged.

- If possible, add failure cases and give appropriate analysis.

**Strengths And Weaknesses:**

Strengths:
- Data Contribution: The consolidation of eight diverse autonomous driving datasets into a unified format, augmented with reasoning annotations, is a practical contribution that enhances model generalisation.
- Efficiency Approach: Shifting from autoregressive token generation to parallel trajectory regression is a logical and effective approach to address the latency bottleneck in VLM-based driving agents.
- Performance: The proposed method demonstrates strong performance on standard benchmarks, outperforming several baselines in both open-loop and closed-loop metrics.

Weaknesses:
1. Originality and Relationship with Prior Work: The novelty of the methodological contribution is limited, particularly concerning the training pipeline and reward design. The proposed training strategy (SFT + RL) and the design of reward functions appear to be heavily derived from $AutoDrive-R^2$. The physical rewards in Reasoning-VLA are essentially simplified versions of the Spatial, Dynamics, and Temporal Smoothness rewards presented in $AutoDrive-R^2$. Simply removing the text-format reward (necessitated by the architectural shift to a regression head) does not constitute a significant methodological innovation. I strongly recommend that the authors remove the fourth core contribution listed in the Introduction chapter.

2. Soundness of Experimental Comparisons: The claims about inference efficiency are not adequately supported by fair comparisons. While the paper emphasises "efficiency" throughout the text, it fails to report or discuss inference speed comparisons with existing state-of-the-art VLA models. The comparison provided in the Appendix is restricted to a generic Qwen2.5-VL-7B model in autoregressive mode. This only demonstrates the inherent advantage of parallel decoding over sequential generation. Furthermore, the reported inference speed of 89ms is achieved on an NVIDIA H200. Although the authors did not provide external comparisons, it is worth noting that existing models like DriveVLM achieve ~410ms on much weaker edge devices (NVIDIA Orin). Given the large computational gap between an H200 and an Orin, the proposed method might be slower than current SOTA methods when evaluated on comparable edge hardware.

3. Presentation and Clarity
- Confusing Terminology: The paper distinguishes the proposed method from "autoregression-based" approaches while simultaneously describing ARM as employing a "regression-based strategy". While I understand the distinction lies between sequential token generation (classification) and direct continuous value prediction (regression), the similarity in terminology can be confusing. I suggest clarifying this distinction early in the methodology section, explicitly contrasting "discrete token generation" with "continuous coordinate regression" to avoid ambiguity.
- Redundant Text: The three research questions listed at the beginning of the Experiments section are generic and superfluous. Standard experimental sections in this field inherently address performance comparisons, component contributions, and efficiency. Explicitly listing them consumes valuable space without adding informational content. I strongly suggest removing these questions to condense the text.

---

> ### Author Rebuttal · Authors · 2026-03-26
>
> We sincerely appreciate the time and effort you have dedicated to reviewing our manuscript. Your insightful comments have significantly improved the quality of our research. Below, we address your concerns point by point.
>
> Weakness 1:
>
> Our work began following the release of QWen2.5VL in March 2025. Inspired by reasoning VLMs, we designed this paper around the SFT+RL strategy. Due to the extensive effort required for dataset cleaning, annotation, and numerous open-loop and closed-loop experiments, we were unable to meet the ICLR deadline.
>
> **Trajectory Reward**
>
> Inspired by mathematical reasoning in VLMs, we use Euclidean distance as the basis of our trajectory reward, but it differs from $AutoDrive-R^2$ in several key aspects:
>
> Our equation is:
>
> $
> r_{\text{traj}} =1- \frac{1}{N} \sum_{i=1}^{N} \gamma^i\left( \alpha(x^i - x_{\text{gt}}^i)^2 + \beta(y^i - y_{\text{gt}}^i)^2 \right)
> $
>
> $AutoDrive-R^2$:
>
> $
> r_p =\frac{1}{N} \sum_{i=1}^{N} \left( (x^i - x_{\text{gt}}^i)^2 + (y^i - y_{\text{gt}}^i)^2 \right)
> $
> 1. Our reward increases as the Euclidean distance between predicted $(x, y)$ and ground truth $(x_{\text{gt}}, y_{\text{gt}})$ decreases. Thus, we use 1 - euclidean-distance. In contrast, $AutoDrive-R^2$'s reward decreases as the Euclidean distance decreases, which can paradoxically incentivize the model to predict positions far from the ground truth in order to achieve a higher reward.
>
> 2. When we conduct our experiments, we observed that prediction errors increase for future time steps (1s, 2s, 3s). Simply averaging these errors treats all steps equally is not appropriate, so we introduce a discount factor $\gamma$ to assign higher weight to near-future predictions and lower weight to distant-future predictions.
>
> 3. Vehicles have limited lateral (y-axis) movement compared to longitudinal (x-axis) movement. Experiments also show that using standard Euclidean distance causes slower convergence along the x-axis direction. To balance this, we apply weighting factors $\alpha$ and $\beta$.
>
> **Steering Reward**
>
> The steering reward is designed based on vehicle kinematics, where the maximum steering angle is physically limited to 40 degrees.
> \begin{equation}
> \frac{( y^j - y^{j-1})}{(x^j - x^{j-1})} < tan(40) = 0.84
> \end{equation}
>
> Predictions satisfying this constraint (0.84) are rewarded with 1; otherwise, 0. The reward increases as the difference between the predicted and true values decreases. Unlike $AutoDrive-R^2$, which can reward excessive steering, our formulation prevents training collapse.
>
> **Acceleration Reward**
>
> Vehicle acceleration is limited to $0.6g$ ($6~\text{m/s}^2$). Predictions within this limit are rewarded with 1; otherwise, 0. $AutoDrive-R^2$ uses velocity and temporal smoothness rewards, which can still incentivize large deviations from ground truth.
>
> To sum up, in autonomous driving, RL rewards can target position, acceleration, and steering. As suggested, since $AutoDrive-R^2$ employs similar rewards, we will remove the fourth core contribution listed in the Introduction.
>
> Weakness 2 and Key Question 1:
>
> DriveVLM used models smaller than 4B (e.g., Gemma-2B, Qwen4B, Qwen1.8B). Since DriveVLM is not open-sourced, we selected comparable models (Qwen4B and DriveLM) for benchmarking.
>
> |Model|Speed(s)|
> |--|--|
> |Qwen4B|0.224|
> |DriveLM|2.072|
> |AutoVLA|2.329|
> |$AutoDrive-R^2$|5.516|
> |Reasoning-VLA-7B|0.081|
> As seen from the table, our method is efficiency in reference.
>
> Weakness 3:
>
> We apologize for the unclear expression and have added terminology clarification in the methodology section:
>
> Autoregression-based: Generates discrete tokens sequentially (similar to classification).
>
> Regression-based Strategy: Predicts continuous values using a regression strategy.
>
> We also thank the reviewer for pointing out the redundant text regarding the three research questions at the beginning of the Experiments section; this has been removed from the paper.
>
> Key Question 2:
>
> We plan to upload our dataset to Hugging Face. To preserve double-blind review, it has not yet been released. We will provide access in the discussion phase if required.
>
> Limitations:
>
> Thank you for your suggestion. We have added a separate discussion section.
>
> Ambiguity in Trajectory Regression: Without explicit high-level navigation commands (e.g., ``turn left/right''), the model relies on regression objectives that cannot fully capture multi-modal futures. In ambiguous scenarios, this can lead to mode averaging and unrealistic trajectories. Incorporating language-based navigation instructions can mitigate this.
>
> Dependence on Teacher Model Quality: Training relies on chain-of-thought supervision from a large VLM. Errors or biases in the teacher can propagate to our model, which is critical in safety-sensitive applications like autonomous driving.
>
> Failure Cases: In complex urban intersections with occlusions, our model may predict invalid trajectories, reflecting mode averaging and incomplete understanding of the 3D environment.

---

> > ### Author Rebuttal · Reviewer_jNHv · 2026-04-02
> >
> > All questions and advice are replied/accepted by the authors. The score is adjusted to 4.

---

> > > ### Author Response · Authors · 2026-04-02
> > >
> > > Dear Reviewer, We truly appreciate your time and effort in reviewing our manuscripts and offering your valuable feedback. We thank you for allowing us to revise, which greatly moved us. We will carefully revise our manuscript.
> > >
> > > Once again, thank you for raising score.
> > >
> > > Wish you all the best and great success with your paper!
> > >
> > > Warm regards
> > >
> > > Authors

---

### Official Review · Reviewer_gg6N · 2026-02-16

**Soundness:** 3
**Presentation:** 3
**Significance:** 3
**Originality:** 3
**Overall Recommendation:** 4
**Confidence:** 4

**Summary:**

Reasoning-VLA is a vision-language-action model for autonomous driving that builds on a reasoning-capable VLM backbone (Qwen2.5-VL) and bridges it to trajectory/action prediction via parallel action decoding. The method introduces learnable action queries (instead of token-by-token generation) with implicit spatial guidance during initialization, followed by refinement modules that interact with intermediate VLM states to improve trajectory accuracy and stability. Training includes supervised fine-tuning on a unified multi-dataset reasoning set and RL fine-tuning using verifiable, rule-based rewards that incorporate trajectory deviation and basic vehicle dynamics constraints.

**Compliance With Llm Reviewing Policy:**

Affirmed.

**Final Justification:**

My concerns have been resolved. Overall, the paper is well-structured and presents clear contributions. After also considering the feedback from other reviewers, I will maintain a positive score.

**Key Questions For Authors:**

See weaknesses.

**Limitations:**

No. The paper should discuss failure modes (OOD scenes, rare events), safety risks from erroneous trajectories, dataset bias, and misuse (e.g., over-trusting model outputs).

**Strengths And Weaknesses:**

# Strengths
1. Parallel action decoding via learnable action queries is a clear architectural departure from autoregressive action-token generation, with explicit spatial guidance and refinement to improve action precision and efficiency.
2. Targets a practical bottleneck in VLA driving—slow sequential decoding—by producing action chunks in one step, aiming for faster inference suitable for online driving.
3. RL fine-tuning uses explicit, verifiable reward designs, including trajectory alignment and simplified vehicle-dynamics constraints, which improves interpretability of optimization signals.

# Weaknesses
1. The RL rewards encode hand-crafted thresholds/constraints (e.g., steering-angle criterion) that may be dataset- or scenario-specific, and it is unclear how sensitive results are to these choices.

Overall, this is a meaningful and technically solid piece of work that addresses an important efficiency bottleneck in reasoning-based VLA driving. The system is well-engineered and empirically validated. However, while the integration is coherent, the individual components appear largely incremental, and none of them stands out as fundamentally novel on its own.

---

> ### Author Rebuttal · Authors · 2026-03-30
>
> Dear Reviewer,
>
> We sincerely appreciate the time and effort you dedicated to reviewing our manuscript. Your insightful comments have significantly improved the quality of our work. We have carefully addressed your concerns in detail and hope our responses are satisfactory.
>
> **1. Rebuttal for weaknesses: "The RL rewards encode hand-crafted thresholds/constraints (e.g., steering-angle criterion) that may be dataset- or scenario-specific".**
>
> In our paper, the physical trajectory reward, steering constraint reward, and acceleration reward are designed with distinct logic.
>
> **Physical Trajectory Reward**
>
> During RL fine-tuning, inspired by math problem-solving in reasoning VLMs, we found that using simple Euclidean distance for trajectory positions did not yield satisfactory performance. Visualization of dataset trajectories revealed significant variance along the x-axis but minimal variance along the y-axis. Additionally, trajectory errors increased over time (1s, 2s, 3s).
>
> To address this, we analyzed human driving behavior and the distribution of trajectories across our 8 datasets. Based on these statistics, we designed an advanced physical trajectory reward incorporating weighting factors $\alpha$ and $\beta$ and a discount factor $\gamma$ to better reflect realistic driving dynamics.
>
> **Steering Constraint Reward**
>
> Vehicles have kinematic limitations, with a maximum steering angle of 40 degrees. For two adjacent trajectory points, this imposes:
> \begin{equation}
> \frac{y^j - y^{j-1}}{x^j - x^{j-1}} < \tan(40^\circ) \approx 0.84.
> \end{equation}
> Our steering constraint reward is therefore based on the physical design of vehicles.
>
> **Acceleration Reward**
>
> Vehicle acceleration is constrained by dynamics, with a maximum acceleration limited to $0.6g$. The acceleration reward reflects this fundamental physical constraint.
>
> **2. Rebuttal for weaknesses:"it is unclear how sensitive results are to these choices.".**
>
>
> We conducted ablation studies to evaluate the influence of each reward. Open-loop experimental results on the nuScenes dataset (Reasoning-VLA-7B RL-finetuned on our datasets) are shown below:
>
> |Model|L2(1s)|L2(2s)|L2(3s)|L2(avg)|
> |--|--|--|--|--|
> |w/o $r_{traj}$|0.15|0.31|0.64|0.37|
> |w/o $r_{steer}$|0.07|0.27|0.52|0.29|
> |w/o $r_{acc}$|0.06|0.26|0.50|0.27|
> |Reasoning-VLA-7B|0.05|0.20|0.44|0.23|
>
>
> The results indicate that removing the physical trajectory reward significantly degrades performance (average L2 increase of 0.14). Removing the steering or acceleration rewards causes smaller decreases (0.06 and 0.04, respectively). This demonstrates that our method is highly sensitive to the physical trajectory reward and moderately sensitive to the steering and acceleration rewards.
>
> **3. Rebuttal for Limitations:**
>
> We appreciate the reviewer highlighting the limitations of our work. We have added the following discussion:
>
> **Failure Modes (OOD Scenes and Rare Events)**
>
> Autonomous driving systems remain vulnerable to out-of-distribution (OOD) scenarios and rare events (e.g., unusual pedestrian behavior, extreme weather, atypical road structures). In our closed-loop experiments, the simulator includes scenarios such as objects inverting from the side, vehicles suddenly merging in front, and sensor degradation. Table 2 shows that our model achieves lower collision rates compared to competing models but still experiences performance degradation relative to in-distribution scenarios.
>
>
> **Safety Risks from Erroneous Trajectories**
>
>
> As illustrated in Figure 4 (Argoverse visualization), our method initially produces erroneous trajectories. However, when the ego vehicle approaches a traffic cone, the model corrects its trajectory to avoid the obstacle. This demonstrates the model’s self-correcting ability, though large trajectory errors can still cause failures, explaining the remaining collisions in closed-loop tests.
>
>
> **Dataset Bias**
>
> Our dataset comprises 8 different base datasets, which introduces inherent biases. For example, MapillaryV2 contains countryside scenarios, IDD includes village scenarios, and nuScenes features nighttime scenes. Our model predicts both common urban daytime driving behavior and under-represented rare or hazardous conditions. The strong reasoning ability of the large vision-language backbone allows our model to mitigate dataset bias, as demonstrated by zero-shot cross-dataset generalization experiments.
>
> **Misuse and Over-Trust in Model Outputs**
>
> Our model is not a standalone decision-making system but a component of a larger safety-critical pipeline. We emphasize the need for human- and system-level oversight, redundancy (e.g., multiple perception/planning modules), and human-in-the-loop validation during early deployment to mitigate risks associated with over-trust.

---

> > ### Author Rebuttal · Reviewer_gg6N · 2026-04-02
> >
> > My concerns have been resolved. Overall, the paper is well-structured and presents clear contributions. After also considering the feedback from other reviewers, I will maintain a positive score.

---

> > > ### Author Response · Authors · 2026-04-03
> > >
> > > Dear Reviewer, We truly appreciate your time and effort in reviewing our manuscripts and offering your valuable feedback. We thank you for allowing us to revise, which greatly moved us. We will carefully revise our manuscript.
> > >
> > > Wish you all the best and great success with your paper!
> > >
> > > Warm regards
> > >
> > > Authors

---

### Official Review · Reviewer_JTkE · 2026-02-24

**Soundness:** 3
**Presentation:** 3
**Significance:** 3
**Originality:** 2
**Overall Recommendation:** 4
**Confidence:** 3

**Summary:**

The paper presents Reasoning-VLA, a VLA framework for autonomous driving that enables one-step parallel trajectory generation via learnable action queries interacting with a reasoning-enhanced vision–language model. The method includes spatially guided initialization, an action refinement module, and SFT+RL training on a unified multi-dataset reasoning benchmark. Experiments show strong open- and closed-loop performance, cross-dataset generalization, and substantial inference speed improvements over VLA baselines.

**Compliance With Llm Reviewing Policy:**

Affirmed.

**Final Justification:**

The rebuttal addresses my main concerns in a meaningful way. The additional single-dataset comparison helps better isolate the architectural contribution from data-scale effects, and the reward ablation strengthens the empirical support for the RL design. Although I still think the paper could provide stronger analysis of novelty and efficiency, the response makes the work more convincing overall. I am therefore updating my recommendation to 4.

**Key Questions For Authors:**

1. Could the authors clarify whether there is any theoretical or representational advantage of the proposed design over standard DETR-style query decoding applied to trajectory regression? Does the parallel action query formulation provide a different inductive bias compared to conventional decoder queries, or are there optimization or stability benefits that can be formally characterized?
2. Since the main results are obtained under the unified multi-dataset training setup, it remains unclear whether the performance improvements stem primarily from the proposed action-query architecture or from large-scale multi-dataset exposure. Could the authors provide strictly controlled comparisons, like training all methods on a single dataset under identical splits, to better isolate the architectural contribution?
3. Although substantial inference speed improvements are reported, the efficiency claim is largely empirical. Could the authors provide a more systematic complexity analysis (e.g., theoretical decoding cost comparison), full end-to-end latency measurements (including visual encoding and refinement), and scaling behavior under different batch sizes or KV-cache settings to more rigorously support the parallel decoding advantage?
4. The inclusion of zero-shot experiments is valuable; however, it would be helpful to more explicitly quantify the generalization gap relative to in-domain performance. Could the authors report relative performance drops and, if possible, closed-loop zero-shot results to better characterize cross-domain robustness?

**Limitations:**

1. Limited Clarity on Architectural Novelty. The learnable action query design is closely related to existing DETR-style query paradigms. The paper does not provide sufficient theoretical analysis or controlled comparisons to clearly demonstrate a fundamentally new learning formulation or representational advantage.

2. Difficulty in Isolating Architectural Gains from Data Scale Effects. Since the strongest results are obtained under unified multi-dataset training, it remains unclear how much of the improvement comes from the proposed architecture versus large-scale multi-domain data exposure. More strictly controlled data comparisons would strengthen the claims.

3. Efficiency Claims Lack Formal Characterization. Although significant empirical speedups are reported, the efficiency advantage is not supported by theoretical complexity analysis, end-to-end latency breakdown, or scaling studies, limiting the rigor of the parallel decoding claim.

4. Limited Analysis of RL Design and Generalization Boundaries. The RL stage relies on manually designed rewards without detailed sensitivity analysis or stability evaluation. Additionally, while zero-shot results are provided, the generalization gap and potential failure modes are not systematically analyzed.

**Strengths And Weaknesses:**

Strengths:
1. The experimental section is thorough, covering both open-loop and closed-loop evaluations across multiple time horizons and metrics. The improvements are consistently observed, and the inclusion of closed-loop NeuroNCAP evaluation significantly strengthens the practical relevance of the work.
2. The construction of a unified dataset from eight public autonomous driving benchmarks represents a substantial engineering effort. The standardized reasoning-based data format and multi-domain training setup contribute to relatively stable cross-dataset performance.

Weaknesses:
1. The core architectural design appears closely related to existing transformer query paradigms such as DETR-style learnable queries. While the adaptation to trajectory prediction is practically meaningful, the manuscript does not clearly articulate a fundamentally new learning formulation or provide analysis demonstrating representational or optimization advantages over standard query-based decoding mechanisms.
2. Although the paper claims improved efficiency through parallel action generation, this claim is primarily empirical and lacks formal characterization. There is no explicit computational complexity comparison, latency analysis, or scaling discussion to substantiate the efficiency gains. Moreover, it is difficult to disentangle whether the observed performance gains are attributable to the proposed architectural design or are primarily a consequence of large-scale multi-dataset training. The paper would benefit from controlled experiments isolating these two factors.
3. The reinforcement learning stage adopts GRPO with manually designed reward functions (trajectory, steering, acceleration), but the paper does not provide deeper theoretical or empirical analysis of how each reward component affects generalization or stability. The RL contribution appears more as an adaptation of existing techniques rather than a novel advancement.

Overall, this is clearly a substantial engineering effort. The unified dataset construction, large-scale training setup, and comprehensive evaluations suggest that the authors have invested significant time and care into the work. The system is thoughtfully built and empirically strong. And the paper would benefit from deeper theoretical grounding and more controlled analysis to clearly separate architectural contributions from data and training effects. Strengthening these aspects would make the work even more convincing.

---

> ### Author Rebuttal · Authors · 2026-03-30
>
> We sincerely thank the reviewer. We address each concern below.
>
> Weakness 1, key question 1 and limitation 1:
>
> We apologize for the lack of clarity. Our learnable action queries are specifically designed for
>
> action-expert-based VLA. Unlike conventional designs (e.g., DETR), we unify them into a single
>
> intra-action query attention mechanism, thereby reducing computational fragmentation and improving optimization stability.
>
> Let:
> \[\begin{aligned}
> Q_{learn} &\in \mathbb{R}^{L \times d}: \text{learnable action query},\\
> H &\in \mathbb{R}^{N \times d}: \text{kv-cache hidden states from VLM},\\
> S &=
> \begin{bmatrix}
> {Q}_{learn} \\
> H
> \end{bmatrix}
> \in \mathbb{R}^{(L+N) \times d}: \quad \text{concatenated sequence},\\
> W_Q, W_K, W_V &\in \mathbb{R}^{d \times d}: \text{projection matrices},\\
> M &\in \(-\infty,0\)^{L \times (L+N)}: \text{action attention mask}.
> \end{aligned}\]
>
> The Q, K, V are:
> \[
> \begin{aligned}
> Q = Q_{learn} W_Q \in \mathbb{R}^{L \times d}, \\
> K = S W_K \in \mathbb{R}^{(L+N) \times d},\\
> V = S W_V \in \mathbb{R}^{(L+N) \times d},\\
> A = \frac{Q K^\top}{\sqrt{d}} \in \mathbb{R}^{L \times (L+N)}.\\
> \end{aligned}
> \]
>
> **Action Attention Masking**
>
> The action mask is merged the self-atten and cross-atten parts:
> $
> M = \big[ M_{self} \mid M_{cross} \big],
> $
> where $M_{self} \in \{-\infty,0\}^{L\times L}$ controls intra-action query attention,
>
> and $M_{cross} \in \(-\infty,0\)^{L\times N}$ controls query to kv-cache attention.
>
> Masked attention weights:
> $
> \tilde{A} = A + M,\qquad
> \alpha = \mathrm{softmax}(\tilde{A}) \in \mathbb{R}^{L \times (L+N)}.
> $
>
> Then the output is:
> $O = \alpha V \in \mathbb{R}^{L \times d}.$
>
> By block decomposition:
> $
> \alpha = \big[ \alpha_{self} \mid \alpha_{cross} \big],\qquad
> V = \begin{bmatrix} V_Q \\ V_H \end{bmatrix},
> $
>
> the output naturally decomposes into self-atten and cross-atten:
>
> $
> O = \underbrace{\alpha_{self}V_Q}_{Self-Atten}+
> $
>
> $
> \underbrace{\alpha_{cross}V_H}_{Cross-Atten}
> $
>
> This unified formulation integrates self- and cross-atten within a single operation, simplifying
>
> computation and stabilizing training dynamics.
>
>
> Weakness 2, key question 2\&3 and limitation 2\&3:
>
> **Autoregressive Decoding**
>
> Given a token sequence $x_{1:T}$, autoregressive decoding factorizes the joint probability as:
>
> $P(x_{1:T}) = \prod_{t=1}^{T} P(x_t \mid x_{<t})$
>
> At inference, tokens are generated sequentially:
> $x_t = \arg\max_{x} P(x \mid x_{<t})$
>
> Complexity: Let $C$ denote per-step compute cost, then total latency:
>
> $\mathcal{T}_{AR} = \sum^{T}_t C = \mathcal{O}(T C)$
>
> **Our Parallel Decoding**
>
> Parallel decoding removes causal dependency:
>
> $P(x_{1:T}) \approx \prod_{t=1}^{T} P(x_t \mid \mathbf{z})$
>
> where $\mathbf{z}$ is a VLA latent representation.
>
> Decoding is performed simultaneously:
>
> $x_{1:T} = \arg\max_{x_{1:T}} \prod_{t=1}^{T} P(x_t \mid \mathbf{z})$
>
> Complexity: All tokens computed in parallel:
>
> $\mathcal{T}_{PD} = \mathcal{O}(C)$
>
> **Mathematical Comparison**
>
> $
> \text{AR: } x_t \perp x_{>t} \mid x_{<t}, \quad
> \text{PD: } x_t \perp x_{\neq t} \mid \mathbf{z}
> $
>
> Error Propagation:
> For AR, define per-step error $\epsilon_t$, then cumulative error is
>
> $\mathbb{E}[\text{Error}_{AR}] \ge \sum^{T}_t \epsilon_t$
>
> For parallel decoding, no recursive amplification:
>
> $\mathbb{E}[\text{Error}_{PD}] = \sum^{T}_t \epsilon_t $
>
> Latency Speedup:
>
> $
> \mathcal{T}_\text{AR}  \div
> $
>
> $
> \mathcal{T}_{PD} = \mathcal{O}(T)
> $
>
> Parallel decoding achieves:
>
> $\mathcal{T}_{PD} $
> <<
>
>  $\mathcal{T}_{AR}$
>
> while trading off exact dependency modeling:
>
> $
> P_{PD}(x_{1:T}) \neq P_{AR}(x_{1:T})
> $
>
> Thus, parallel decoding provides linear speedup while avoiding error accumulation. The
>
> experimental results reported in Table 9 of our paper empirically validate this phenomenon.
>
> In addition, as shown in Tables7 and 8, when trained solely on the nuScenes dataset (same with
>
> prior methods), our approach achieves an average L2 error of 0.25 in open-loop evaluation,
>
> outperforming existing methods by 0.05. Similarly, in closed-loop evaluation, our method improves
>
> the NeuroNCAP score by 0.10. These results demonstrate the superiority of our architectural
>
> design over existing approaches.
> Due to hardware constraints, all experiments are conducted with a batch size of 1.
>
> Weakness 3 and limitation 4:
>
> We further evaluate the impact of different reward components (open-loop on nuScenes):
> |Model|L2(avg)|
> |--|--|
> |w/o $r_{traj}$|0.37|
> |w/o $r_{steer}$|0.29|
> |w/o $r_{acc}$|0.27|
> |Ours|0.23|
>
> Removing the trajectory reward leads to the significant degradation (+0.14 L2), while steering and
>
>  acceleration rewards result in moderate degradation (+0.06 and +0.04, respectively). This
>
> indicates that trajectory consistency is the most critical factor in our framework.
>
> Key question 4:
>
> For closed-loop zero-shot evaluation, we adopt the NAVSIM simulator, which provides stable
>
> closed-loop environments. As reported in Table 6, our method consistently outperforms existing
>
> approaches in terms of final PDMS score.

---

> > ### Author Rebuttal · Reviewer_JTkE · 2026-04-01
> >
> > The rebuttal addresses my main concerns in a meaningful way. The additional single-dataset comparison helps better isolate the architectural contribution from data-scale effects, and the reward ablation strengthens the empirical support for the RL design. Although I still think the paper could provide stronger analysis of novelty and efficiency, the response makes the work more convincing overall. I am therefore updating my recommendation to 4.

---

> > > ### Author Response · Authors · 2026-04-02
> > >
> > > Dear Reviewer,
> > >
> > > Thank you once again for your valuable comments and constructive suggestions. Your feedback has been extremely helpful in improving the quality of our manuscript. Your positive evaluation is very encouraging, and we will continue to revise and refine the manuscript accordingly.
> > >
> > > Wishing you great success with your paper!
> > >
> > > Wish you all the best.
> > >
> > > Warm regards
> > >
> > > Authors

---

### Decision · Program_Chairs · 2026-04-30

**Decision:**

Accept (regular)

**Comment:**

This paper proposes Reasoning-VLA, a VLA which improves upon existing VLM-based frameworks by optimising the output mechanism to accelerate inference speed. Authors introduce learnable action queries instead of token-by-token generation, with implicit spatial guidance during initialization. Additionally, the authors integrate eight mainstream autonomous driving datasets to construct a unified dataset, including Chain-of-Thought annotations generated by a VLM to facilitate effective VLA training.

Reviewers praised the paper for its data contribution, efficient shift from autoregressive token sequences to direct regression, explicit and verifiable reward designs, and strong performance on benchmarks.

Reviewers identified concerns related to novelty, reward handcrafting, and experimental analysis, but all were addressed sufficiently in the rebuttal such that reviewers returned positive impressions. Remaining concerns include limited clarity of architectural novelty, and difficulty in analyzing whether gains are structural from architecture improvements or data scaling, which may be worth continued investigation by the authors in continued work.

From the contributions of the paper in both a new architectural paradigm for VLA driving and engineering a unified dataset, I recommend acceptance.